# Updated Clinical Evidence on the Role of Adipokines and Breast Cancer: A Review

**DOI:** 10.3390/cancers15051572

**Published:** 2023-03-03

**Authors:** Georgios-Ioannis Verras, Levan Tchabashvili, David-Dimitris Chlorogiannis, Francesk Mulita, Maria-Ioanna Argentou

**Affiliations:** 1Breast Unit, Department of General Surgery, General University Hospital of Patras, 26504 Rio, Greece; 2Department of D/I Radiology, Patras General Hospital, 26332 Patras, Greece

**Keywords:** breast cancer, adipokines, adiponectine, leptin, resistin, visfatin

## Abstract

**Simple Summary:**

Breast cancer is currently one of the most common types of cancer and the number one cause of cancer-related deaths in women worldwide. Despite significant advances involving cancer research in cancer biology, targeted treatments, and novel surgical approaches, breast cancer poses a constant, prominent challenge. In order to combat this reality, novel biomarkers and treatment targets are constantly on the rise. One of the known risk factors and survival predictors of breast cancer is obesity and obesity-related hormonal changes. The main effectors of said hormonal changes are a group of fatty tissue-related molecules, adipokines. Adipokines have many known and intertwined mechanisms of actions, many of which are known to enable carcinogenesis within the breast tissue. This review aims to summarize all available evidence of relationships between adipokines and the development of breast cancer, in order to emphasize their potential roles as novel biomarkers, predictive indicators, and possible future therapeutic targets of breast cancer.

**Abstract:**

With the recent leaps in medicine, the landscape of our knowledge regarding adipose tissue has changed dramatically: it is now widely regarded as a fully functional endocrine organ. In addition, evidence from observational studies has linked the pathogenesis of diseases like breast cancer with adipose tissue and mainly with the adipokines that are secreted in its microenvironment, with the catalog continuously expanding. Examples include leptin, visfatin, resistin, osteopontin, and more. This review aims to encapsulate the current clinical evidence concerning major adipokines and their link with breast cancer oncogenesis. Overall, there have been numerous meta-analyses that contribute to the current clinical evidence, however more targeted larger-scale clinical studies are still expected to solidify their clinical utility in BC prognosis and reliability as follow-up markers.

## 1. Introduction

The days in which adipose tissue (AT) was considered a fat-storage and thermoregulatory organ are long over. The extra endocrine nature of adipose tissue was solidified with the discovery of a biological compound in the face of leptin and its function in 1994 [1]. Years of consequent hormone discoveries have shed light on its true endocrine nature and thus replaced this simplistic belief with an understanding of adipose tissue as a complex organ that is involved in many physiological pathways from inflammation to carcinogenesis and especially breast cancer.

Breast cancer (BC) is an umbrella term encompassing tumors arising from each different cell type of the breast tissue, with most of them being adenocarcinomas. The current epidemiological data places BC as the second most common malignancy in women in the United States and as the second most lethal type of female cancer. It is hypothesized that one in eight women in the US will eventually develop breast cancer. Its incidence has been slowly increasing, which is mainly attributed to a drop in fertility rates and, most importantly, to the Western lifestyle of fatty, rich, and processed food consumption and lack of exercise, leading to obesity [2].

Obesity is a pandemic in itself. However it is also closely correlated with an increased risk for the development of breast cancer; it also leads to worse patient outcomes when present [3]. Obesity has been positively correlated with worse prognoses [4] and is considered an independent risk factor for disease progression in major diseases like dyslipidemia, hypertension, cardiovascular disease (CVD), diabetes mellitus (DM) type 2, stroke, chronic kidney disease, and carcinogenesis [5]. Furthermore, the International Agency for Research on Cancer (IARC) concluded that there is plenty of available data to support the direct link [6] between excessive fat disposition and tumor development for multiple types of cancer like gastric cardia, esophageal adenocarcinoma, liver, gallbladder, pancreas, kidney, colon, and rectum, while low BMI was found to be protective against thyroid, gastric cardia, pancreas, liver, gallbladder, ovary, multiple myeloma, and meningioma. In addition, an inverse correlation between adult-type post-menopausal breast cancer risk and waist circumference has been well established [7].

Multiple mechanisms have been proposed to explain this dose-dependent relationship of fat disposition and oncogenesis, for example, inflammatory and immunologic pathways or epigenetic changes that result in DNA damage, alteration, and eventual malignant transformation. Recent studies have focused on the role of adipokines, hormones that are secreted by the adipose tissue, on oncogenesis that in an obesity-state can act directly on metabolic pathways such as Janus kinase–signal transducer and activator of transcription (JAK-STAT) or phosphoinositide 3-kinase (PI3K) and pathways or via tumor microenvironment alteration [8]. Furthermore, over ten adipokines have been associated with breast cancer, with their number steadily increasing [9]. The increase of the majority of circulating adipokines like adiponectin, leptin, resistin, visfatin, osteopontin, apelin, and lipocalin has been linked with breast cancer, while on the contrary reduced circulating levels of the adipokines adiponectin and iridine (also known as adipo-mycin) has been shown to bear a protective role against it.

Obesity is a well-known alternator of tissue microenvironments, It has long been identified as a central link in the chain of pathophysiological events leading to carcinogenesis. All the above-mentioned intracellular pathways are activated following extracellular changes in the microenvironment via crosstalk mechanisms that transfer extracellular stimuli via a molecular cascade. One of the most prominent elements of epigenetic modification in cancer is epigenetics involving mitochondrial DNA (mitoepigenetics) that are affected primarily by oxidative stress [10]. Mitochondria also appear to play a central role in the production of metabolites that cause epigenetic modification of nuclear DNA that at times causes oncogenic mutations [11]. One of the key processes of oncogenic effects is DNA hypermethylation, which results in the inactivation of tumor-suppressor genes within the cell [11]. Examples within breast cancer include DNMT-1 gene disinhibition and SIRT4 loss, mediated by hypermethylation, which promotes breast cancer self-renewal [11]. Oxidative stress is known to be mediated by hormonal changes in obesity patients, in particular by upregulation of pro-inflammatory cytokines caused by altered levels of adipokines that in turn cause the production of more adipokines, creating a molecular loop [12]. It has been demonstrated that mismanagement of reactive oxygen species by cancer cells can result in altered mitoepigenetics, which lead to defects in the mitochondrial genome of breast adipose cells that are vital to tumor progression and infiltration [10]. Microenvironment alteration in the form of increased free ROS is directly associated with increased triglyceride deposition within adipose cells, which induces an inflammatory cellular response and endoplasmic reticulum stress [13]. In addition to carcinogenesis, obesity and adipokine-mediated microenvironment alterations have been found to be correlated with worse survival outcomes, with a significant proportion of obese patients being diagnosed with triple-negative breast carcinoma, a subtype known for inferior survival and increased resistance to chemotherapy [13]. Overall, a deeper understanding of the heterogenous role of adipokines in the breast cancer tumor microenvironment will aid the evidence-based practices of breast cancer treatment by revealing new markers for disease progression and potential therapeutic targets.

Herein, we will review the current clinical evidence of the role of certain adipokines in cancer pathogenesis with an emphasis on meta-analyses and synopsize the underlying mechanisms of certain adipokines that contribute to the neoplastic development of breast cancer. A visual summary of current insights on the role of adipokines in breast cancer can be seen in Figure 1.

## 2. Adipose Tissue Microenvironment and Breast Cancer Cells

The role of adipose tissue in breast cancer development was based on two observations. Firstly, from the plethora of data that have established the relationship of obesity and post-menopausal breast cancer development [14] and progression [15,16]. Secondly, adipose tissue represents the largest part of the microenvironment that nurtures breast cancer cells, irrespective of the patient’s BMI [17]. In obese states, the adipose tissue is in a continuous low-inflammatory state in which the fat cells secrete hormones that can directly alter metabolic pathways and promote mutations [18] or indirectly make changes in the microenvironment by recruiting inflammatory cells. The recruited immune cells in the tumor microenvironment not only fail to exhibit cancer suppression activities, they also paradoxically promote “immune-evasion” phenomena by promoting different receptor expression in tumor cells that ultimately make the immune cells unable to identify and destroy them [19].

The peritumoral adipose tissue, though interactions with cancer cells, dedifferentiate into pre-adipocytes or differentiate into cancer associated adipocytes (CAA) and continue to secrete adipokines that promote an infiltrating phenotype [20]. Moreover, it is theorized that CAAs, in order to support breast cancer cells, undergo reprogramming of their intracellular metabolic pathways through their direct interaction. Through the dynamic exchange of metabolites like fatty acids, cancer cells act like parasites and take advantage of this energy source through β-oxidation to meet their increased metabolic demands.

Indeed, chemokines like chemokine (C–C motif) ligand 5 (CCL5), chemokine (C–C motif) ligand 2 (CCL2), interleukin-1β (IL-1β), tumor necrosis factor-alpha (TNF-α), and vascular endothelial growth factor (VEGF), secreted by CAAs, orchestrate tumor invasion of the basement membrane and distant metastasis through the circulatory system.

Although changes in tumor microenvironments are often thought of as a local process, adiposity-induced changes can also stem from distant adipose mass. Systemic low-grade inflammation is a cornerstone in the relationship between obesity and carcinogenesis, a relationship best described by the actions of released adipokines (studied here), interleukin 6 (IL-6) and Tumor Necrosis Factor α (TNF-α) [21]. Adipose-tissue-derived factors engaged in the regulation of inflammatory response and the management of oxidative stress are also produced in more than just breast adipose cells, and can be found in visceral and subcutaneous adipose tissue [15]. Current evidence also suggest a degree of crosstalk between different adipose tissue subtypes, mediated by exosomes that can promote intracellular communication via the transportation of molecules that are pro-inflammatory and pro-fibrotic in nature [15]. Extracellular vehicles containing thrombospondin-5 (TSP-5) that promotes epithelial-to-mesenchymal transition of breast adipocytes have been identified as related to progression and infiltration of breast carcinoma [22]. Obesity, as a systemic condition, exerts oncogenic effects on breast tissue; cancer-associated fibroblasts that are stimulated from adipokines provide an altered extracellular matrix that favors the development of malignant adipocytes and has been shown to activate when co-cultured with adipose cells taken from subcutaneous tissue [16,21]. Clinical studies have also confirmed that visceral adiposity is closely related to breast cancer outcomes, with a 2021 study clearly indicating a correlation between the metabolically hyperactive visceral adipose tissue commonly found in obese patients and worse recurrence-free survival rates [23].

## 3. Adipokines

Adipokines (also known as adipocytokines) is a vast group of heterogenous soluble factors produced by adipose tissue that function in different pathways involving metabolism, inflammation, and vascular homeostasis (Table 1). The catalogue includes over 600 identified proteins [24], with the best known being adiponectin, leptin, resistin, visfatin, osteopontin, IL-6, NF-κB, etc. Adipokines interact with and activate different pathways that contribute to the hallmarks of breast cancer since they express respective receptors for the interaction.

### 3.1. Adiponectin

Adiponectin is the protein hormone of the C1q/TNF family. It is comprised of 244 amino acids and encoded by the ADIPOQ gene located in chromosome 3q27. It is mainly secreted by white adipose tissue and primarily has energy homeostasis and anti-inflammatory effects. It also promotes insulin sensitivity and cell proliferation. It exerts its function by binding to the AdipoR1 (mainly expressed in skeletal tissue and endothelial cells) and AdipoR2 receptors (mainly expressed in the liver) [25].

Recent studies have shown that CAA is a known effector of decreased adiponectin secretion in humans. Adiponectin has also been shown to act as a protective factor against tumor progression through its most prominent receptors: AdipoR1 and AdipoR2. Their interaction inhibits the growth and invasion of cancerous cells, including those of breast cancer. It is also known to induce cell apoptosis by enabling Adenosine 5′-monophosphate (AMP)-activated protein kinase (AMPK) signaling and inhibiting PI3K/AKT signaling [9,25,26]. Adiponectin also seems to act in a contradictory manner to leptin. The adiponectin/leptin ratio is often used in literature to describe their interaction. Obesity development is characterized by the mis-differentiation of adipocytes. A result of this mis-differentiation is the induction of hypoxia-induced factor 1 (HIF-1) that in turn stimulates the expression of leptin and downregulates the expression of adiponectin [9]. In other words, the adiponectin/leptin ratio is decreased in the adipose tissue of obese people. The same receptor also promotes the activation and translocation of LKB1/STE20- related adaptor proteins and sc and scaffolding mouse 25 protein (MO25) from the nucleus all the way to the cytoplasm, which causes the subsequent activation of AMPK and at the same time the inhibition of MAPK, WNT-beta-cantenin, NFkB, and more [27].

Recent epidemiological data have shown adiponectin to have an inverse relationship to obesity-related cancers [28] (Table 2). Low serum adiponectin levels have been correlated with a high risk of breast cancer [29], while high serum levels may act protectively against it. This finding, however, has been subject to controversy since several individual studies did not find an increased risk. Ever since, adiponectin became a hot topic of extensive research and recent meta-analyses have tried to clarify the true association while also taking into account the ethnicity and dietary heterogeneity of different ethnic populations. As such Gu et al. [30] and Yu et al. [31] pooled the results of 31 and 27 studies, respectively, in order to obtain a reliable sample size. The results of these studies confirmed the inverse relationship of serum levels of adiponectin and breast cancer in pre-menopausal and post-menopausal women, with the included studies exhibiting a high heterogeneity. Subsequent subgroup analysis stratified by ethnic groups revealed that the association was stronger in women of Asian heritage than the Caucasian group. A different meta-analysis by Yoon et al. [32] studied the circulating levels of different adipokines in different cancer types and reported that the cancer type with the stronger connection with low serum adiponectin levels was breast cancer.

Among the other reported actions of adiponectin, some of the most prominent are its protective function against obesity-related diseases such as metabolic syndrome, cardiovascular disease, type II diabetes, and malignancies. Hypoadiponectinemia has been correlated with insulin resistance, diabetes, and cardiovascular risk, in addition to malignancy development. A better understanding of the link between adiponectin levels and cancer cell proliferation and metastatic potential is needed, and it could provide insights to potential therapeutic targets. Adiponectin has been studied in vitro where it was found to suppress cell proliferation, invasion, and migration in estrogen-deficient breast cancer cell lines [33]. The literature suggests a somewhat inconsistent relationship between adiponectin and the development of breast cancer. A meta-analysis by Liu et al. found that adiponectin levels were inversely associated with breast cancer with an OR of 0.838 [34]. It must be noted, however, that these results were highly influenced by certain individual studies that caused a high degree of heterogeneity, and it was only after their removal from the analysis that a marginally significant result was obtained. Likewise, regarding the menopausal status of patients, the relationship seemed to be reversed. Postmenopausal women were more likely to develop breast cancer alongside high levels of adiponectin. Additionally, the study was not able to identify any significant associations between adiponectin and breast cancer in all stratified sub-analyses [35].

### 3.2. Leptin

Leptin is a 146-amino acid type I cytokine and a member of the family of long-chain helical cytokines. It is encoded by the Ob gene located in chromosome 7q31.3. Leptin is predominantly produced in the adipose tissue. It serves as an indirect modulator of fat tissue mass by producing the signal of satiety in the hypothalamus and its production is proportionate to the total adipose tissue mass.

Apart from its endocrine activities, leptin also exhibits various pre-oncogenic mitogenic actions through the LEPR receptor, which is widely expressed in breast cancer cells. Leptin activates PI3K/AKT and JAK/STAT pathways and induces their proliferation [36]. It also inhibits cancer cell apoptosis by inducing the expression of anti-apoptotic genes like bak, bax, and angiogenesis though VEFG production. Even though these proposed mechanisms have been widely accepted, there have been contradicting results regarding leptin’s role in the risk of breast cancer development. A meta-analysis conducted by Niu et al. was one of the first to establish this positive correlation [37]. Moreover, a different team [38] that examined the association of multiple adipokine levels in different cancer types underscored that ER+ cancer patients had significantly higher leptin levels than ER- cases (Table 2). The association between leptin levels and estrogen status in breast cancer has been previously studied as well. Raut et al. [39] proved that the suppression of estrogen receptor signaling downregulates leptin-induced cellular cycle progression and the induction of leptin-controlled autophagy process. Leptin is also known to activate estrogen receptors via transactivation stimulated by leptin through MAP-kinase signaling [40]. In addition, leptin levels were also significantly higher in lymph node metastasis (LNM) positive cases than in LNM negative cases, and considerably higher between post-menopausal and pre-menopausal women.

Contrary to the effects of adiponectin, leptin seems to promote the proliferation and development of breast cancer cells. Production of leptin from the adipose tissue is directly proportional to the total adipose tissue mass of an adult. Moreover, studies have shown an additional leptin production site to be present in the form of fibroblasts associated with cancer [15]. Another interesting mechanism of action that promotes oncogenesis through leptin is its crosstalk with produced estrogens. Estrogen receptors and leptin receptors ObR were found to be coexpressed in malignant breast tissue, as well as cell lines from breast cancer specimens [41]. Other studies have proven that there is an association between leptin levels, estrogen, and progesterone receptor expression in breast cancer patients. Previous works have also proven a physiological mechanism through which estradiol levels upregulate leptin m-RNA expression in adipose tissue and increase leptin and ObR expression in breast cancer cell lines [33,40,41]. Leptin activates ER-alpha through the MAP-Kinase pathway in breast cancer cells, and in the process, reproduces the features of ER-alpha transactivation.

One of the most popular concepts in breast cancer carcinogenesis is the epithelial-to-mesenchymal transition (EMT), which is otherwise a normal physiological process found mainly in wound healing and embryogenesis. EMT is a critical step in carcinogenesis. It represents the turning point of an epithelial cell towards a depolarized, mesenchymal-type cell [27]. As a consequence of the loss of cell polarity, these cells gain cancerous characteristics—survival, loss of adhesion, and potential for invasion—thus adding to the tumor’s overall aggressiveness. A recent study showed that ObR RNA expression and concomitant leptin secretion is found in cancer-associated fibroblasts (CAFs) and proposed that leptin is integral in mediating the crosstalk between potentially cancerous breast cells and CAFs, causing tumor growth and invasive characteristics [41]. The authors describe that, when breast cancer cells were treated with leptin, morphological phenotypical changes occurred that highly resembled fibroblast cells, with increased pseudopodia formation, actin reorganization, and stress fiber formation. The exposure of cells to leptin also altered E-cadherin expression by downregulation and caused the upregulation of mesenchymal markers.

Laboratory studies, however, seem to differentiate from epidemiological studies as far as the role of leptin in breast cancer goes. Subsequent observational studies were not able to verify the risk for breast cancer development in post-menopausal women [42] or presented weak associations [43] A research team from Iran [44] also found no association with cancer in leptin’s gene and leptin’s gene receptor, polymorphism LEP rs7799039 and LEPR rs1137101, respectively. The previously established results were however recently overturned, largely due to a meta-analysis from Pan et al. [44] that presented solid evidence from multiple sources regarding the positive correlation of elevated leptin levels and incidence rates of breast carcinoma, especially in obese post-menopausal women. A subgroup analysis of the same study also identified a stronger association in patients of Asian descent, more specifically from China.

### 3.3. Resistin

Resistin, originally discovered in 2001, is a small 108-amino acid cystine-rich adipokine encoded by the RETN gene in chromosome 19 (19p13.2). Resistin is involved not only in glucose and insulin metabolic pathways (its name stems from resistance to insulin), but also in pro-inflammatory pathways (NF-κB, PI3K) through the activation of its receptor TLR4 (a member of the Toll-Like Receptor family) [45,46]. Thus, it has been found that resistin is secreted not only by adipocytes but also by inflammatory cells.

An insight into the pathophysiology of resistin in the proliferation and invasion of breast cancer cells can be found in the work of Lee et al. [47]. Resistin was found to mediate breast cancer cell invasion through the c-Src pathway that resulted in an intracellular calcium influx, after which cancer cells exhibited invasive properties. Resistin was also found to induce PKC alpha phosphorylization, a pathway known for its ability to induce the invasive properties of cancer cells. One of the most prominent sites of action of the PKC alpha pathway is the increase of vimentin filaments, a process best known for its contribution to the epithelial to mesenchymal transition in carcinogenesis [27,41].

Clinical implications of hyperestistinemia include contribution to diabetes type 2 through increased insulin resistance, obesity, and, from the activation of the pro-inflammatory pathways in obesity, malignancies like colorectal and breast cancer [44,48]. However, in recent years there have been conflicting results regarding the relationship between RETN polymorphisms as a prognostic factor for breast cancer. A meta-analysis that pooled the data from nine individual studies showed an increased risk of breast and colon cancer in patients with the RETN rs1862513 variant, especially Caucasians. In addition, there was no data to support the association of the RETN rs3745367 variant with malignancy. Moreover, its role as a prognostic marker for breast cancer has also been studied with conflicting results. Yoon et al. reported that high levels of serum resistin did not bear statistical significance, with the studies included bearing high heterogeneity [32] (Table 2). Of note is that during sensitivity analysis, the exclusion of one study resulted in a positive association with breast cancer. On the contrary, the results from the study of Gui et al. underlined an increase in the mean concentrations of resistin in the breast cancer group [37]. However, this association was observed in the Asian group, with concomitant no significant difference between the non-Asian and control group.

### 3.4. Visfatin (eNampt)

Visfatin, also known as nicotinamide phosphoribosyl-transferase (NAMPT) or pre-B-cell colony-enhancing factor 1 (PBEF1), originally discovered in 2005, is a large 52kD protein with enzymatic activity. It is encoded in the NAMPT gene located in chromosome 7 (7q22.2). Two forms have been identified, an intracellular form (iNampt) that is involved in NAD salvage pathways, and an extracellular form (eNampt) that is found in many tissues ranging from adipose to heart tissue. It is involved in beta oxidation, inflammatory pathways, and angiogenetic pathways [49].

Visfatin has attracted the attention of the scientific community due to its relationship with obesity-related cancers and, most specifically, post-menopausal breast cancer. Moreover, its hypersecretion is related with worse prognoses. It has been hypothesized that visfatin promotes breast cancer survival through the ABL proto-oncogene 1 (c-Abl) activator of the transcription 3 (STAT3) pathway and via upregulation of the mRNA levels of cyclin D1 and cyclin-dependent kinase 2 (CDK2). Many studies have examined the relation of serum visfatin levels with breast cancer. Of note are the results of a large meta-analysis, which included 27 studies, that indicated that malignant individuals have significantly different elevated serum visfatin levels compared to controlled cohorts [50] (Table 2). Consistent with these results was the recently reported outcome of a study by Hori Ghaneialvar et at [51] in which it was found that the levels of visfatin are different between breast cancer patients and healthy individuals. These results highlight visfatin’s potential future application as a screening and/or follow-up tool. Lastly, a study by Gui et al. found no strong difference in the serum levels of visfatin between subjects with BMI < 25 kg/m^2^ and those with BMI > 25 kg/m^2^.

### 3.5. Lipocalin-2

Lipocalin-2, also known as neutrophil gelatinase-associated lipocalin (NGAL), is a 198 amino acid adipocytokine, member of the lipocalin family. It is encoded by the LCN2 gene located in chromosome 9 (9q34.11). Lipocalin-2, as an acute phase reactant protein, is mainly expressed by neutrophils, with its main role being iron-sequestration and thusthe limiting of bacterial growth, while also elevated in the early stages of acute kidney injury. However, recent studies have reported that it is also secreted by adipose tissue and may contribute to breast cancer progression [52] by inducing EMT [53]. Its role as a biomarker was proposed by the results of a meta-analysis by Wang et al. in which NGAL levels were correlated with breast cancer diagnosis [54,55]. Subsequent studies have strengthened the association between elevated NGAL levels and breast cancer [55,56] and, more specifically, the upregulation of the LCN2 [56,57] with consequent activation of the e-cadherin pathway, starting from the activation of the transmembrane glycoprotein cadherin that is known to stimulate cellular invasion and metastasis properties via tumor-stroma interactions, a process that apparently leads to significantly poor disease outcomes [57] (Table 2). Lastly, silencing of the LCN2 gene was also proposed as a novel therapeutic target via small interference RNA molecules (siRNA). Ginette et al. assessed the biological efficacy of in vitro silencing molecules for inflammatory breast cancer, which resulted in decreased cell proliferation and invasiveness [58].

### 3.6. Chemerin

Chemerin, also known as retinoid acid receptor responder protein 2 (RARRES2), is a small 16-kDa protein. It is encoded by the RARRES2 gene in chromosome 7 (7q36.1). Chemerin has been shown to act as a chemoattractant by acting as a ligand to Chemerin/chemokine-like receptor (CMKLR-1) expressed in immune cells [59], while also exhibiting angiogenetic and proliferative inducing properties. There is a paucity of data with conflicting results concerning its value as a biomarker in cancer research. Some studies have shown that RARRES2 downregulation is associated with poor prognoses in certain cancer types, like hepatocellular carcinoma, mainly due to diminished leukocyte recruitment [60], while also acting as a favorable prognostic marker in lung cancer [61]. Within contrast to breast cancer, a study revealed higher expression of chemerin in malignant tissue in comparison with adjacent normal breast tissue and was associated with poor prognosis [62]. In addition, a study by Song et al. confirmed that elevated serum levels of chemerin (in combination with CA15-3) achieve better diagnostic performances in breast cancer and correlate to aggressive phenotypes [63] (Table 2). Lastly, a team led by Pachynski have proposed novel therapeutic strategies by studying the in vivo properties of chemerin and showing that it suppressed its growth by NK and T cell recruitment within the breast cancer’s tumor microenvironment [64].

Chemerin is most prominently known for its receptor CMKLR1, expressed in dendritic cells and other components of the immune system. Its role in chemotaxis and attraction of immune cells towards the inflammatory response site is well known and hypothesized to contribute to carcinogenesis indirectly through the induction of inflammatory responses that turn the tumor microenvironment into a favorable state for carcinogenesis [64].

### 3.7. Osteopontin (OPN)

Osteopontin, also known as bone/sialoprotein I (BSP-1), is a large adipokine derived from the bone as it was firstly found as part of the normal bone tissue extracellular matrix, but is also expressed in a variety of other tissues from adipocytes to the placenta [65]. It is encoded by the SPP1 gene located in chromosome 4 (4q1322.1). Its role through the activation of integrins is biomineralization, chemotaxis, inflammation, and cell activation.

Osteopontin merits scientific research due to recent observations of its relationship to many diseases, from obesity and diabetes to a variety of cancer types [66] like breast cancer and hepatocellular carcinoma. Indeed, abundant expression of osteopontin has been associated with poor prognosis and low survival. In a meta-analysis by Hao et al. [67] (Table 2), the prognostic value of osteopontin was studied in breast cancer. The final analysis included 1567 breast cancer patients and the results underlined the strong correlation of high osteopontin levels and worse overall mortality. In addition, its splice variant-c expression appeared to be even more significantly associated with worse prognosis, making osteopontin and osteopontin-c candidates for future breast cancer prognostic markers.

**Table 1 cancers-15-01572-t001:** Included Studies.

Meta-Analysis (Study)	Number of Studies	Number of Participants	Study Outcomes
**Adiponectin**			
Gu L et al. Serum adiponectin in breast cancer: A meta-analysis. Medicine (Baltimore) 2018; 97:e11433 [30]	31 eligible studies were included	Βreast cancer group (7388), Control Group (8491)	Lower serum adiponectin levels in BC cases, in pre-menopausal and post-menopausal women, especially among Asians but not in the Caucasian population.
Yu Z, Tang S, Ma H, Duan H, Zeng Y. Association of serum adiponectin with breast cancer: A meta-analysis of 27 case-control studies. Medicine (Baltimore). 2019 Feb;98(6):e14359 [31].	27 eligible studies were included	Βreast cancer group (7176), Control Group (8318)	Serum adiponectin was inversely associated with breast cancer. Decreased serum adiponectin levels in pre-menopausal women, post-menopausal status. In addition, low serum adiponectin levels in Asian women were more likely to be associated with breast cancer risk than in Caucasian women.
Yoon YS, Kwon AR, Lee YK, Oh SW. Circulating adipokines and risk of obesity related cancers: A systematic review and meta-analysis. Obes Res Clin Pract. 2019 Jul-Aug;13(4):329–339 [32]	14 eligible studies were included		By cancer site and type, highest category of adiponectin was associated with decreased risk of breast cancer
**Leptin**			
Pan H, Deng LL, Cui JQ, Shi L, Yang YC, Luo JH, Qin D, Wang L. Association between serum leptin levels and breast cancer risk: An updated systematic review and meta-analysis. Medicine (Baltimore). 2018 Jul;97(27):e11345 [44].	35 eligible studies were included	Brest Cancer Group (6086) vs. Control Group (7158)	Serum leptin levels were related to breast cancer risk as demonstrated by calculations of the overall SMD = 0.46 (95% CI = 0.31–0.60, I2 = 93.5%). A subgroup analysis of BMI identified an association between breast cancer and serum leptin levels in patients who are overweight and obese (overweight: SMD = 0.35, 95% CI = 0.13–0.57, I2 = 88.1%; obesity: SMD = 1.38, 95% CI = 0.64–2.12, I2 = 89.6%). Additionally, menopausal status subgroup analysis revealed a significant association in post-menopausal women (SMD = 0.26, 95% CI = 0.12–0.40, I2 = 77.9%). Furthermore, we identified a significant association between breast cancer and serum leptin levels in Chinese women (SMD = 0.61, 95% CI = 0.44–0.79, I2 = 40.6%).
Yoon YS, Kwon AR, Lee YK, Oh SW. Circulating adipokines and risk of obesity related cancers: A systematic review and meta-analysis. Obes Res Clin Pract. 2019 Jul-Aug;13(4):329–339. [32]	9 eligible studies were included		In the linear dose-response analysis by cancer type, each 5 ng/mL increase in leptin was not significantly associated with BC. Leptin was significantly associated with increased risk of cancer.
Sayad S, Dastgheib SA, Farbod M, Asadian F, Karimi-Zarchi M, Salari S, Shaker SH, Sadeghizadeh-Yazdi J, Neamatzadeh H. Association of PON1, LEP and LEPR Polymorphisms with Susceptibility to Breast Cancer: A Meta-Analysis. Asian Pac J Cancer Prev. 2021 Aug 1;22(8):2323–2334. [43]	12 studies on LEP rs7799039, and 14 studies on LEPR rs1137101 were selected.	3444 cases and 3583 controls on LEP (leptin) rs7799039 polymorphism, 5330 cases and 6188 controls on LEPR(leptin receptor) rs1137101 polymorphism	LEP rs7799039 and LEPR rs1137101 polymorphisms were not associated with an increased risk for breast cancer.
Gui Y, Pan Q, Chen X, Xu S, Luo X, Chen L. The association between obesity related adipokines and risk of breast cancer: a meta-analysis. Oncotarget. 2017 May 13;8(43):75389–75399 [37].	46 eligible studies were included	Brest Cancer Group (6459) vs. Control Group (7155)	ER positive cases had significantly higher leptin levels than ER negative cases, Leptin and TNF-α levels were also significantly higher in lymph node metastasis (LNM) positive cases than in LNM negative cases, Leptin levels were significantly higher among post-menopausal cases than pre-menopausal cases
**Resistin**			
Hashemi M, Bahari G, Tabasi F, Moazeni-Roodi A, Ghavami S. Association between rs1862513 and rs3745367 Genetic Polymorphisms of Resistin and Risk of Cancer: A Meta-Analysis. Asian Pac J Cancer Prev. 2018 Oct 26;19(10):2709–2716. [68]	9 studies were included	1951 cancer patients and 2295 healthy controls were included	The data revealed no correlation between the rs3745367 polymorphism and cancer risk.
Yoon YS, Kwon AR, Lee YK, Oh SW. Circulating adipokines and risk of obesity related cancers: A systematic review and meta-analysis. Obes Res Clin Pract. 2019 Jul-Aug;13(4):329–339 [32].	5 eligible studies were included		The highest vs. lowest meta-analysis showed no relationship between resistin and cancer risk, with high heterogeneity between studies. Exclusion of one study resulted in significance during sensitivity analysis.
Gui Y, Pan Q, Chen X, Xu S, Luo X, Chen L. The association between obesity related adipokines and risk of breast cancer: a meta-analysis. Oncotarget. 2017 May 13;8(43):75389–75399. [37]	6 studies were eligible	Brest Cancer Group (1236) vs. Control Group (1137)	Mean concentrations were higher than in the control group, but among Asian cases and not in non-Asian groups.
**Visfatin (eNampt)**			
Gui Y, Pan Q, Chen X, Xu S, Luo X, Chen L. The association between obesity related adipokines and risk of breast cancer: a meta-analysis. Oncotarget. 2017 May 13;8(43):75389–75399. [37]	3 studies were eligible	Brest Cancer Group (433) vs. Control Group (271)	Mean concentrations of visfatin were higher in cases than controls. There was no significant difference in the levels of visfatin between subjects with BMI > 25 kg/m^2^ and those with BMI < 25 kg/m^2^
Ghaneialvar H, Shiri S, Kenarkoohi A, Fallah Vastani Z, Ahmadi A, Khorshidi A, Khooz R. Comparison of visfatin levels in patients with breast cancer and endometrial cancer with healthy individuals: A systematic review and meta-analysis. Health Sci Rep. 2022 Nov 18;5(6):e895. [51]	10 studies were eligible	Breast Cancer group (260) Control Group (400)	Increased levels in breast cancer.
Mohammadi M, Mianabadi F, Mehrad-Majd H. Circulating visfatin levels and cancers risk: a systematic review and meta-analysis. J Cell Physiol. 2019;234(4):5011–5022. [69]	27 studies were eligible	Breast Cancer group (2693) Control Group (3040)	Metaresults showed a significant higher level of visfatin in patients with cancer than in the controls
**Osteopontin (OPN)**			
Hao C et al. Prognostic Value of Osteopontin Splice Variant-c Expression in Breast Cancers: A Meta-Analysis. Biomed Res Int 2016; 2016: 7310694 [67]	10 time-to event studies were eligible	1567 breast cancer patients	High level OPN indicated a poor outcome in the OS. High level OPN-c appeared to be more significantly associated with poor survival. Our analyses indicated that both OPN and OPN-c could be considered as prognostic markers for breast cancers
**Lipocalin-2 (NGAL)**			
Wang Y et al. Neutrophil gelatinase-associated lipocalin protein as a biomarker in the diagnosis of breast cancer: A meta-analysis. Biomed Rep 2013;1:479–83 [70]	4 studies were eligible	Breast Cancer Group (332). Control Group (142)	ROC curve (AUC) for Breast Cancer diagnosis was 0.90. Sensitivity: 64% (95% CI, 0.59–0.69). Specificity: 87% (95% CI, 0.81–0.92).

**Table 2 cancers-15-01572-t002:** Summary of the role of adipokines in carcinogenesis.

Adipokine	Proposed Implications in Carcinogenesis
Adiponectin	Decreased levels of adiponectin are associated with carcinogenesis. Protective effects through the AdipoR1 and AdipoR2 receptors. Downregulation of HIF-1. Activation of MPK.
Leptin	Activation of JAK/STAT and PI3k/AKT pathways that induce proliferation. Induction of angiogenesis through VEGF stimulation. Contributes to EMT that produces CAFs.
Resistin	Pro-glycolic adipokine that promotes inflammatory response via NF-κB and PI3K pathways.
Visfatin	Activates the ABL proto-oncogene 1 the STAT3 pathway and upregulates mRNA levels of cyclin D1 and CDK2.
Lipocalin-2	Pro-inflammatory response, capable of inducing EMT.
Chemerin	Angiogenic and proliferative properties. Binds to the CMKLR-1 receptor. Increased levels seem to exert anti-oncogenic effects.
Osteopontin	Activation of integrins, pro-inflammatory adipokine. Associated with worse survival rates and mortality.

## 4. Conclusions

With obesity steadily overthrowing cigarettes as the leading preventable risk factor for cancer, it is of utmost importance that we decipher the significant connection of its consequences, like elevated circulating adipokines in a chronic inflammatory state, with breast cancer. While many recent meta-analyses revealed a strong association of high levels of certain adipokines (adiponectin, leptin) with breast cancer, inter-population variability may limit its generalizability as a biomarker for disease prognosis and progression. Further clinical studies are needed to produce more robust results and assess their prognostic reliability and novelty as therapeutic targets.

## Figures and Tables

**Figure 1 cancers-15-01572-f001:**
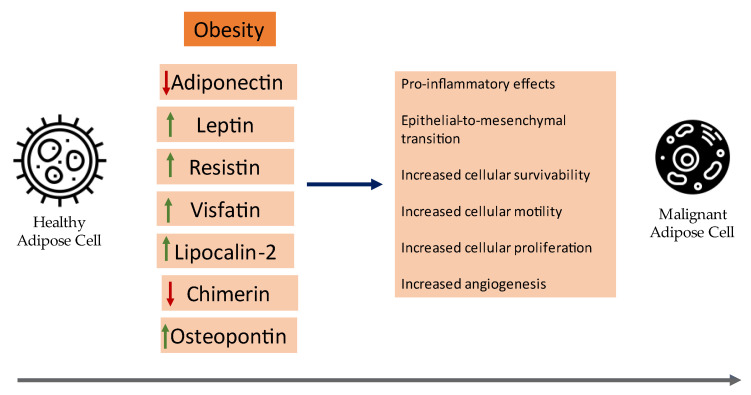
An overview of the role of adipokines in the development of breast cancer.

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
