# Peer review of "Updated Clinical Evidence on the Role of Adipokines and Breast Cancer: A Review"

_cancers, 2023, doi:10.3390/cancers15051572_

Round 1

Reviewer 1 Report

Dear Authors,

The manuscript "Updated Clinical Evidence on the Role of Adipokines and Breast Cancer: A Review" by Verras et al has summarized the role of adipokines in breast cancer. I have just a few suggestions.

1. Some background information or references are missing. In introduction, please add more background information about the relation between obesity, tumor microenvironment and breast cancer. (Please cite:

1. An Epigenetic Role of Mitochondria in Cancer. Cells 2022, 11, 2518. https://doi.org/10.3390/cells11162518

2.     Advances in the Prevention and Treatment of Obesity-Driven Effects in Breast Cancers. Front Oncol. 2022 doi: 10.3389/fonc.2022.820968.

3.     mitochondrial mutations and mitoepigenetics: Focus on regulation of oxidative stress-induced responses in breast cancers. Semin Cancer Biol. 2022 Aug;83:556-569. doi: 10.1016/j.semcancer.2020.09.012.)

2. Please add more figures to show the clear relations between adipokines obesity and breast cancer.

Best,

Author Response

Dear Reviewer,

We would like to extend our gratitude, for taking the time to critically appraise our work, and suggest ways to improve our manuscript. It is the belief of the entirety of the authoring team, that your suggestions and proposed references greatly added to the value of our Review, and have made our overview of the literature complete. Find below a point-for-point answer to your comments.

  • All three of the proposed articles are now cited within our revised text. The authoring team as a whole, felt that indeed, they constituted valuable addittions to examined literature. More background information has been added in the Introduction section (Lines 78-110) as well as the Adipokines section (Lines 135 - 159)
  • A new figure was made from scratch, summarising the role of adipokines in carcinogenesis (Figure 1)

It is our firm belief that our manuscript is now complete, and by taking into account your valuable suggestions, will constitute a significant addition to literature regarding breast cancer carcinogenesis.

Reviewer 2 Report

The manuscript by Verras et al. "Updated Clinical Evidence on the Role of Adipokines and Breast Cancer: A Review " has potential, however, some aspects need it to be improved

Main concern

On the other hand, a paragraph showing the evidence to date and explaining the particular implications of adipose tissue in the breast versus visceral in a physiological and pathophysiological context such as breast cancer is important to include. For example, is there some degree of communication or signaling between both adipose tissues? Unfortunately, the authors only mentioned a brief sentence about that.

Line 59, remove the letter "d" at the end of the paragraph.

From lines 68 to 75, a reference is missing for important topics, such as the 10 adipokines associated with breast cancer, which is the main topic of this review.

From line 83 is missing a reference(s) for the link between obesity and breast cancer progression.

In point 3.1 Adiponectin, from lines 113 to 133, there are no references to support the aspects addressed. In addition, it is suggested to rewrite the information raised from lines 130 to 133.

The paragraph between lines 150 to 157 requires references; in this same when the authors indicate "estrogen deficient breast cancer cells," are they referring to the triple negative breast cancer type or the generation of knockout cell lines?

The paragraph between lines 168 to 171 is repeated between lines 123 to 126; please corroborate if it important to repeat this information.

In point 3.2 Leptin, how, in the authors' opinion, is explained what is stated in the paragraph between lines 185 to 187 regarding the higher levels of leptin in ER-positive tumors less invasive compared to ER-negative breast cancer tumors? What is expressed in this point also requires having the corresponding references to what is indicated.

The paragraph between lines 191-204 also requires references to support what is indicated there. Also, clarify if ERa should be understood as ER-alpha (ER-l), if so, it is suggested to use this last nomenclature

The paragraphs between lines 205-214 and 214-218 also require references to support what is stated there. In line 217, also explain if what is indicated refers to a downregulation effect on E-cadherin levels.

What is expressed in the paragraph between lines 219-227 requires revision since the relevance of including said information is not understood.

In point 3.3 Resistance, there is a missing reference at the end of the first paragraph, line 234. One or more references that support what is indicated is necessary. Again, PKCa refers to PKC alpha ??

In point 3.5 Lipocalin-2, correct the use of italic font from lines 283 to 314. Also, explain what "e-cadherin pathway" refers to in line 293 of this part.

In point 3.6, Chemerin, there are no references to support what is indicated in the paragraph between lines 315-320.

The authors' analysis of the evidence reviewed and summarized in Table 2 would be more useful if presented in one figure.

Some important articles suggested to be included:

·       Lyu X, Zhang Q, Fares HM, Wang Y, Han Y, Sun L. Contribution of adipocytes in the tumor microenvironment to breast cancer metabolism. Cancer Lett. 2022 May 28;534:215616. doi: 10.1016/j.canlet.2022.215616. Epub 2022 Mar 3. PMID: 35248624.

·       Tsankof A, Tziomalos K. Adiponectin: A player in the pathogenesis of hormone-dependent cancers. Front Endocrinol (Lausanne). 2022 Oct 6;13:1018515. doi: 10.3389/fendo.2022.1018515. PMID: 36277714; PMCID: PMC9582436.

·       Nehme R, Diab-Assaf M, Decombat C, Delort L, Caldefie-Chezet F. Targeting Adiponectin in Breast Cancer. Biomedicines. 2022 Nov 17;10(11):2958. doi: 10.3390/biomedicines10112958. PMID: 36428526; PMCID: PMC9687473.

·       Rajput PK, Sharma JR, Yadav UCS. Cellular and molecular insights into the roles of visfatin in breast cancer cells plasticity programs. Life Sci. 2022 Sep 1;304:120706. doi: 10.1016/j.lfs.2022.120706. Epub 2022 Jun 9. PMID: 35691376.

·       Zhou X, Zhang J, Lv W, Zhao C, Xia Y, Wu Y, Zhang Q. The pleiotropic roles of adipocyte secretome in remodeling breast cancer. J Exp Clin Cancer Res. 2022 Jun 14;41(1):203. doi: 10.1186/s13046-022-02408-z. PMID: 35701840; PMCID: PMC9199207.

·       Tewari S, Vargas R, Reizes O. The impact of obesity and adipokines on breast and gynecologic malignancies. Ann N Y Acad Sci. 2022 Dec;1518(1):131-150. doi: 10.1111/nyas.14916. Epub 2022 Oct 27. PMID: 36302117.

·        Hillers-Ziemer LE, Kuziel G, Williams AE, Moore BN, Arendt LM. Breast cancer microenvironment and obesity: challenges for therapy. Cancer Metastasis Rev. 2022 Sep;41(3):627-647. doi: 10.1007/s10555-022-10031-9. Epub 2022 Apr 18. PMID: 35435599; PMCID: PMC9470689.

 Minors revision:

In some paragraphs, for example, Gui et al., Without indicating the year or reference number, in others the author et al [xx] are displayed, review.

Author Response

Dear Reviewer,

We would like to extend our gratitude, for taking the time to critically appraise our work, and suggest ways to improve our manuscript. It is the belief of the entirety of the authoring team, that your suggestions and proposed references greatly added to the value of our Review, and have made our overview of the literature complete. Find below a point-for-point answer to your comments.

  • All of the points suggesting reference addition have been taken into account and relevant references have been added to all of the relevant points mentioned within your review. In addition, the sum of the suggested articles made a very welcome addition to our cited literature.
  • Within the paragraph between ex. lines 150-157, we have clarified the type of cells.
  • In point 3.2 regarding Leptin we have expanded our discussion of the role leptin plays in carcinogenesis, and better explained the phenomenon with appropriate references.
  • The paragraph in lines 219-227 has been revised and made clearer for the reader.
  • A new figure was made from scratch, summarising the role of adipokines in carcinogenesis (Figure 1), summarizing what is stated in Table 2.

It is our firm belief that our manuscript is now complete, and by taking into account your valuable suggestions will constitute a significant addition to the literature regarding breast cancer carcinogenesis and its relation with obesity hormones.

Reviewer 3 Report

The paper is of interest to clinicians and oncologists, because it focuse on relevant additional an behavioural factors of carcinogenesis. Obesity is sometimes more important than cigarette smoking in causing cancer development. Regarding this, sometimes it's more relevant to avoid obesity and other factors creating inflammation processes in tissues which at the end can be the cause of cancer development. The author gives an overview about the knowlege of today.

Author Response

Dear Reviewer,

We would like to extend our gratitude, for taking the time to critically appraise our work, and suggest ways to improve our manuscript. The authoring team is deeply grateful for your positive comments and would like to thank you for being a part of our manuscript. Please find below attached the final version of our article.

It is our firm belief that our manuscript is now complete, and will constitute a significant addition to literature regarding breast cancer carcinogenesis.

Round 2

Reviewer 1 Report

Strongly suggest for publication.

Reviewer 2 Report

Dear Authors:

I appreciate that you considered and incorporated most of the suggestions. My only concern is with the quality and self-explanation of the figure that explains the role of adipokines dysregulation in breast cancer.